# The PreQ-20 TRIAL: A prospective cohort study of the oncologic safety, quality of life and cosmetic outcomes of patients undergoing prepectoral breast reconstruction

**Benigno Acea-Nebril, Alejandra García-Novoa**[ID]\*, **Lourdes García Jiménez**[ID]

Breast Unit, Department of Surgery, University Hospital Complex A Coruña, A Coruña, Spain

\* mag_1406@hotmail.com

## Abstract

### Background

Mastectomy currently constitutes a necessary surgical procedure in the oncologic setting and in the context of high risk. Prepectoral breast reconstruction (PBR) has been proposed as a surgical alternative to retropectoral techniques by providing less postoperative morbidity and a better cosmetic result. However, there is a lack of prospective studies that have evaluated its safety and patient-reported satisfaction.

### Methods

We conducted a prospective cohort study to assess the safety, quality of life and cosmetic sequelae of PBR in women with breast cancer and high risk. The study's main objective is to assess the safety of PBR in terms of postsurgical complications and the feasibility of reconstruction (loss of implants). The secondary objectives are to evaluate oncologic safety (local relapses, residual glandular tissue) and to identify factors related to quality of life and cosmetic sequelae. The evaluation of residual tissue will be conducted by MRI 12 to 18 months after the surgery, and the quality-of-life assessment will be performed using the Breast-Q questionnaire. An initial patient evaluation will be conducted 12–18 months after the surgery, and a second evaluation will be performed at 5 years. The estimated sample size is 81 patients.

### Discussion

The PreQ-20 study will analyze the impact of PBR on 3 separate measures: safety, quality of life and cosmetic sequelae. Unlike other studies that analyzed these three measures jointly for women with breast cancer and high risk, this study will individualize the results for these 2 patient groups. This differentiation is necessary from the methodological point of view, given that the 2 patient groups have separate clinical and emotional implications. The assessment of these groups will focus on the following aspects: postoperative complications, local relapses, evaluation of residual glandular tissue and incidence rate of primary

**Data Availability Statement:** The data underlying the results will be present in the studywill be available from clinicaltrials.gov.

**Funding:** The authors received no specific funding for this work.

**Competing interests:** the authors have declared that no competing interest exist.

**Abbreviations:** PBR, prepectoral breast reconstruction; SSM, skin-sparing mastectomy; NSM, nipple (and skin)-preserving mastectomy; RBP, retropectoral breast reconstruction; RGT, residual glandular tissue; MRI, magnetic resonance imaging; RRM, risk-reducing mastectomy; RR, risk reduction.

tumors in the same, the cosmetic sequelae and the satisfaction and the quality-of-life assessment by the patients.

## Trial registration

ClinicalTrials.gov: NCT04642508.

## Background

In recent years, mastectomy techniques have evolved towards less aggressive procedures with increased skin preservation, skin-sparing mastectomy (SSM), and skin and nipple-sparing mastectomy (NSM). Reconstructive procedures have changed the placement of implants from the retropectoral position towards the prepectoral location in an attempt to decrease postoperative pain, prevent animation deformity and improve the refilling of the lower pole. Various studies [1–3] published in the past 5 years have shown that the combination of SSM/NSM and prepectoral breast reconstruction (PBR) ensures good results both in the oncologic context and in reducing risk (RR). The meta-analysis by Li et al. [4] analyzed 16 studies that compared retropectoral breast reconstruction (RBP) with PBR and found no significant differences in the safety of the two procedures (in terms of seroma, implant loss, infection, skin necrosis), although PBR showed a lower incidence of capsular contracture, less pain and higher scores on the Breast-Q questionnaire compared with RBP.

Despite these technical improvements, there are still a number of unknowns regarding various clinical aspects related to PBR such as safety, quality of life and the progression of the cosmetic results. First, from the point of view of safety, the rate of postoperative complications needs to be evaluated, as well as local relapses in patients with breast cancer and primary tumors in women with high risk. There are some studies [5] that have analyzed this aspect, however, the level of evidence for these studies was low. Another aspect related to the safety of SSM and NSM is the presence of residual glandular tissue (RGT) because it can facilitate a local relapse in patients with breast cancer or promote the onset of a primary carcinoma, both in patients with cancer and those at high risk. Various studies have analyzed the presence of RGT after mastectomy [6–16]. None of these studies have reported the rate of RGT or the time interval between mastectomy and the MRI study. The identification of this RGT during follow-up facilitates the joint decision-making process with the patient, either for their specific follow-up or extirpation.

The patient-reported satisfaction and quality of life after PBR have been analyzed by several authors; however, only 3 studies have evaluated them prospectively [17–19]. The main limitations of these studies are the lack of differentiation in the results between patients with breast cancer and RRM, the scarce follow-up and the absence of a preoperative assessment in one of the studies [19]. Most of the women with breast reconstruction experienced changes and deterioration of their breast during the follow-up. There are no available prospective studies that have analyzed these changes during the follow-up for RBP or PBR. Some retrospective studies detect a very low incidence of complications, cosmetic deterioration and replacement of implants [20, 21]. In contrast, prospective studies have detected a higher rate of cosmetic sequelae after PBR [18, 19]. These studies highlighted the low breast satisfaction of patients with rippling, especially those who underwent a prepectoral procedure.

Although at this time SSM/NSM with PBR constitutes a necessary surgical procedure in the oncologic setting and in the context of high risk, there is a lack of prospective studies that have

evaluated their oncologic safety and patient-reported quality of life. Therefore, the objective of this study will be to assess prospectively and with long-term follow-up the safety of sparing mastectomies with PBR. To do this, we will analyze the safety of the technique in terms of implant loss, polyurethane safety, and surgical complications; oncological safety in terms of RGT, local relapses and new tumors; and the resilience of the reconstruction, assessing cosmetic deterioration, the need for new surgeries and long-term patient satisfaction.

## Research hypothesis

Prepectoral breast reconstruction with polyurethane implant after mastectomy is a safe procedure.

## Study objectives

**Main objective.** To evaluate the safety of prepectoral reconstruction in terms of postsurgical complications and the feasibility of reconstruction (implant loss).
Secondary objectives:

1. To evaluate the safety of sparing mastectomy through the identification of RGT, primary carcinomas and breast cancer relapses (patients operated on for breast cancer).

2. To assess satisfaction and quality of life after prepectoral reconstruction in women with breast cancer and those with high risk.

3. To identify factors related to satisfaction and quality of life after prepectoral reconstruction.

4. To evaluate the cosmetic sequelae after prepectoral reconstruction in women with breast cancer and with high risk.

5. To identify factors related to the cosmetic sequelae after prepectoral reconstruction.

## Study design

Prospective single arm cohort study to evaluate the safety, quality of life and cosmetic sequelae of breast reconstruction using prepectoral implantation in women with breast cancer and those at high risk. The safety of the technique is assessed in terms of loss of implants.

## Methods

### Study setting

The study will be conduct in the Breast Unit of the University Hospital Complex of A Coruña, Spain.

### Eligibility criteria

We will include all older women operated on in the Breast Unit through SSM/NSM (unilateral or bilateral) and immediate reconstruction with prepectoral implantation. The study population will include 2 patient groups:

1. Women with breast carcinoma. This group consists of patients with a histological diagnosis of in situ or infiltrating breast carcinoma, either primary or metachronic, which requires mastectomy as the surgical treatment.

2. Women at high risk for breast cancer. This group consists of women evaluated at a high-risk consultation and whose RRM has been approved by the tumor committee of the Breast

Unit of University Hospital Complex of A Coruña. For this procedure, the patients should meet at least one of the following criteria:

○ hereditary breast and ovarian cancer syndrome, either by demonstrated genetic mutation or through their family history.

○ histological diagnosis of high-risk lesions (atypical hyperplasia, ductal carcinoma in situ, lobular carcinoma in situ) associated with family history.

○ high-risk criteria (genetic, histological, family) during the breast cancer follow-up.

## Exclusion criteria

The following clinical conditions were excluded from the study:

- Inability to undergo magnetic resonance imaging during the diagnosis and follow-up.

- Inability to fill out the BREAST-Q questionnaire.

- Unwillingness by the patient to participate in the study.

- Breast sarcomas.

- PBR using expanders.

- Prepectoral reconstruction with mesh

## Interventions

**Mastectomy technique.** The breast removal will be performed using a mastectomy adapted to the breast and optimizing the preservation of the breast's original elements (inframammary fold, skin envelope, fat transitions, nipple-areolar complex) according to each patient's anatomical and oncologic possibilities. To this end, we will employ Carlson type 1, 2, 3 and 4 SSM [22] or NSM through an inframammary incision or vertical pattern.

*Reconstruction technique*. The reconstruction will be performed by placing a silicone implant coated with polyurethane foam (Microthane[TM], POLYTECH Health & Aesthetics Dieburg, Germany. https://polytech-health-aesthetics.com) in the prepectoral position.

The authors confirm that all ongoing and related trials for this drug/intervention are registered.

## Modification and adherence

Does not apply to the type of study.

## Concomitant care

24-hour intravenous antibiotic prophylaxis is performed in all patients. Usually cefazolin. In patients allergic to penicillin, ciprofloxacin should be used.

## Outcomes

Primary outcome measures.

1. Incidence of implant loss (Feasibility of reconstruction). To evaluate the feasibility of prepectoral reconstruction in terms of implant loss the first year after surgery, also during oncological treatment (example: chemotherapy) (Unit of mesure: number of implant loss).

Implant loss is defined as the need to remove the prosthesis for any cause such as strusion, infection or necrosis.

Secondary outcome measures.

1. Incidence of surgical complications—reoperations (Safety). To determine the global incidence of surgical complications, as well as the causes of complications in terms of: postoperative bleeding, skin necrosis, necrosis of the nipple areola complex, seroma and infection.

   - Postoperative bleeding is defined as the appearance of any amount of bleeding that modifies the course of the first 7 postoperative days. Among these modifications we find: the need to prolong admission due to blood drainage and the need for reoperation due to bleeding or hematoma. Superficial ecchymoses that do not modify the immediate postoperative period are not included

   - Skin necrosis is defined as the appearance of areas of skin without vascularization that condition cell death. Necrosis that appeared during the first 3 months after surgery that conditioned some action by the surgeon are included.

   - Within the necrosis of the nipple areola complex, only deep necroses that required some surgical interventions are included.

   - Seroma is defined as the accumulation of periprosthetic fluid that required maintaining the drain for more than 10 days or placing a new one (ultrasound-guided or at operating room).

   - An infection is defined as the need for antibiotics in the first 30 postoperative days. The antibiotic prophylaxis used (Cefazolin the first 24 hours) is excluded.

2. Incidence of residual glandular tissue (Safety). To evaluate the safety of sparing mastectomy through the identification of residual glandular tissue through a magnetic resonance one year after surgery.

3. Incidence of breast cancer relapse (oncological safety). To evaluate the safety of sparing mastectomy through the identification relapses in the same breast during the follow up.

4. Incidence of new breast cancer (safety of risk reducing mastectomy). To evaluate the safety of risk reducing sparing mastectomy in high risk for breast cancer patients through the identification of new breast cancer.

5. Quality of life and satisfaction after a mastectomy with prepectoral reconstruction, using the Breast-Q questionnaire. To assess satisfaction and quality of life after prepectoral reconstruction in women with breast cancer and those with high risk through Breast-Q questionnaire. Also, to identify factors related to satisfaction and quality of life after prepectoral reconstruction (e.g. marital status, psychological illnesses).

6. Incidence of cosmetic sequelae. To evaluate the cosmetic sequelae after prepectoral reconstruction in women with breast cancer and with high risk and Identify factors related to the cosmetic sequelae after prepectoral reconstruction. The investigators will employ the Clough classification, where type I sequelae identified the patients with breast asymmetry but no deformity, type II sequelae were defined by the presence of a deformity that could be corrected using a breast-conserving procedure, and type III sequelae identified those women whose breast showed a deformity or painful fibrosis that could only be solved through mastectomy. To determine the sequelae, the criteria of the breast surgeon will be used, documented through photos.

**Table 1. Schedule for patient assessment in the PreQ-20 study.**

| | STUDY PERIOD | | | | | |
|---|---|---|---|---|---|---|
| | Enrolment | Allocation | Post-allocation | | | Close-out |
| TIMEPOINT** | $-t_1$ | 0 | $t_1$<br>3 months | $t_2$<br>1 year | $t_3$<br>5 years | $t_x$ |
| **ENROLMENT:** | | | | | | |
| **Eligibility screen** | X | | | | | |
| **Informed consent** | X | | | | | |
| **MRI+ Picture + Breast Qpre** | | X | | | | |
| **Allocation/Inclusion** | | X | | | | |
| **INTERVENTIONS:** | | | | | | |
| *[Mastectomy and Prepectoral reconstruction]* | | | X | | | |
| *[Annual Control -MRI+ BreastQ post]* | | | | X | | |
| *[Long Term Follow up + BreastQ post]* | | | | | X | |
| **ASSESSMENTS:** | | | | | | |
| *[List baseline variables]* | X | X | | | | |
| *[List outcome variables]* | | | X | X | X | X |
| *[List other data variables]* | | | X | X | X | X |

MRI: magnetic resonance image

## Participants' timeline (see Table 1 and Fig 1)

**Preoperative assessment.** All patients will be assessed by a surgeon of the unit who will indicate SSM/NSM and assess its feasibility for each patient. Similarly, the decision for the mastectomy will be made in consensus with the multidisciplinary committee. Before the surgery, the patients will undergo a mammography and magnetic resonance imaging to confirm the tumor size and rule out multifocality/multicentricity, as well as an evaluation of the distribution of glandular tissue and transitions between the breast and chest wall.

**Breast magnetic resonance imaging.** From the clinical standpoint, this diagnostic test is necessary for the follow-up of patients with breast reconstruction for the early diagnosis of relapses and deterioration of the breast implant. This study will employ the first magnetic resonance imaging during the postoperative period (between 12 and 18 months) to assess the residual glandular tissue following the mastectomy.

**Breast-Q questionnaire.** The Breast-Q questionnaire is aimed at evaluating patient-reported satisfaction and quality of life through the use of breast reconstruction modules. The questionnaires are the intellectual property of the University of Columbia (New York, USA), which freely allows their use for clinical research. The questionnaire is presented in 2 formats: the preoperative format, which is delivered to patients before the surgery, and the postoperative format, which is delivered to them 12–18 months after the surgery. Likewise, we will conduct a second postoperative assessment at 5 years of the surgery. The final score for each module will be calculated according to the Mapi Research Trust criteria [23] and will range from 0 to 100 (the higher the score, the greater the satisfaction and wellbeing).

**Images.** The evaluation of the cosmetic sequelae requires taking photographs of the patient's chest (from the suprasternal notch to navel). Photographs will be taken prior to the surgery (frontal, right and left lateral), then again at 12–18 months and finally at 5 years of the surgery.

**Follow-up.** The initial follow-up of the patients will be performed by the surgeon. The immediate complications will be assessed during the first 24–48 h in the hospital

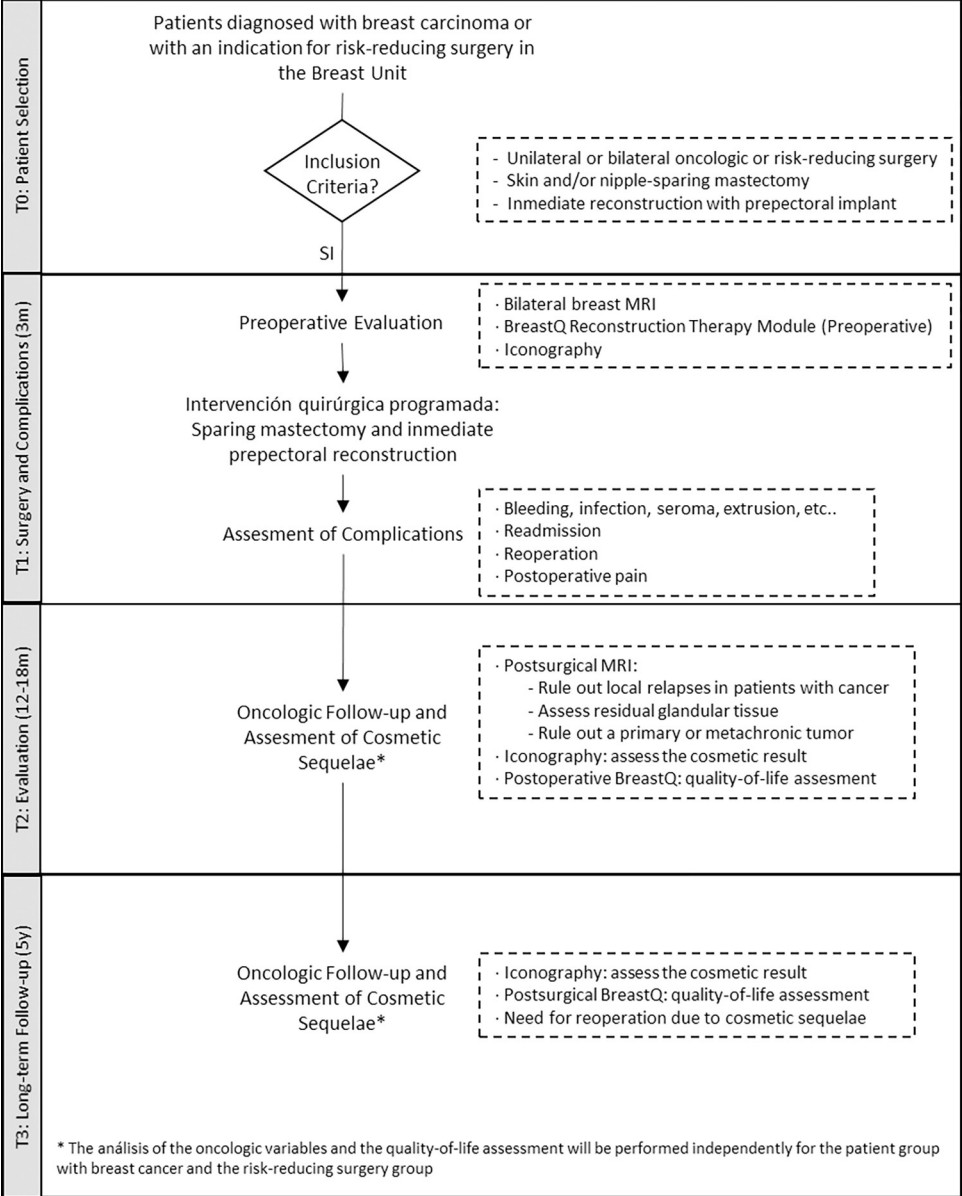

**Fig 1. Methodology algorithm for the PreQ-20 study.**

(postoperative pain and bleeding). The patients will subsequently be followed in the doctor's Breast Unit office. The first visits will be held at 1 week and at 15 days after the surgery. Visits will subsequently be scheduled at 1 and 3 months of the surgery. If there are any complications and/or incidences, an earlier visit will be scheduled.

The patients will visit the Medical Oncology and Radiation Oncology office to complete their adjuvant therapy. The oncology control tests will be performed according to the Breast Unit's protocol, this includes a clinical review (surgeon or medical oncologist) every 3–6 months for the first 5 years. In these reviews, a directed physical examination and the corresponding imaging tests will be performed annually. During these consultations, data will be collected regarding local and systemic relapses.

The women at high risk will undergo annual check-ups in the Breast Unit office (surgeon). All study patients will be evaluated at 12–18 months after the surgery to undergo a breast MRI, imaging and to deliver the postoperative Breast-Q questionnaire. Those patients with RGT will be evaluated by surgeon. The decision to remove this tissue will be made between the patient and the surgeon, considering the amount of residual tissue, the vital status of the patient and the risk of a relapse or new tumor. Women with RGT who do not undergo a reoperation will be followed up with an annual ultrasound and / or MRI.

A second assessment will subsequently be conducted at 5 years during which new images will be taken and the postoperative Breast-Q questionnaire will be delivered.

## Sample size

According to the meta-analysis by Wagner et al. [1] and the iBRA study [21], the incidence rate of implant loss is 3.3% and 9%, respectively. Our study will therefore assume that a 5% loss of implants would be acceptable and will use this value for calculating the sample size. Therefore, with a 95% confidence level (95% CI), an accuracy of 5% and assuming a 10% of potential losses during follow-up, we estimated that the necessary sample size will be 81 patients.

## Recruitment

Recruitment will be done by the surgeons / gynecologists of the breast unit. Every woman diagnosed with breast cancer is referred for surgery. In it, the surgeon will identify the patients who meet the inclusion criteria.

Recruitment began when the Hospital's ethics committee approved the study. therefore, the registration and publication of the protocol has been carried out later.

## Data collection and management

The data sets generated and/or analyzed during the present study will be collected in our center's Breast Unit database and will not be publicly available due to the data protection law and our center's standard protocol. However, the datasets might be available through the corresponding author.

The following variables will be collected for all patients:

- Epidemiological variables: age, weight (kg), height (m), muscle mass index (kg/m$^2$), smoking habit (yes/no), suprasternal notch-nipple distance (cm)and assessment of high risk.

- Healthcare variables: length of stay (days), surgical time (minutes), complications, diagnosis to surgery delay (days), surgery to first adjuvant therapy delay (days), readmission (yes/no), reoperation (yes/no), postsurgical pain at 24–48h and in the first week, measured with the visual analog scale (1 to 10).

- Surgical variables: type of mastectomy (Carlson Classification), surgical specimen weight (g), implant type (volume in cc) and lymph node staging (Sentinel lymph node biopsy or axillary lymph node dissection).

- Histological variables: histological type, nuclear grade, final grade, lymphovascular invasion, intraductal component, lymph node involvement and number of lymph nodes affected.

- Oncologic variables: primary systemic therapy, TNM, final staging, chemotherapy agent, radiotherapy, antihormonal therapy and biological therapy.

- Follow-up variables: residual tissue, residual tissue location, tumor recurrence, tumor recurrence location, deformity, deformity location and asymmetry.

- Satisfaction and quality of life variables (using the Breast-Q questionnaire): breast satisfaction, satisfaction with the information received, satisfaction with the surgeon, satisfaction with the nurses, satisfaction with the team, social wellbeing, sexual wellbeing and physical wellbeing.

### Retention

The assessment and follow-up of the participants included in the study was carried out in the consultations usually scheduled for follow-up and treatment. In this way, patients do not usually leave the study.

### Statistical analysis

We will conduct a descriptive analysis of the variables included in the study. All quantitative variables will be expressed with the mean, standard deviation and corresponding 95% confidence interval. Comparison of proportions: The differences between the various qualitative variables will be found using Fisher's exact test or the chi-squared test ($\chi 2$). The qualitative variables will be expressed in proportions and respective confidence intervals. The differences between the quantitative variables will be analyzed using Student's t-test for independent groups. If the conditions of the t-test are not verified, we will use the Mann-Whitney U test. We will use Kaplan-Meier curves and compare them using the log-rank test to determine those variables related to locoregional relapse and the incidence rate of new tumors, as well as to assess cosmetic deterioration. The statistical analysis will be performed using version 24 of the statistical program IBM SPSS and Epidat 4.1.

Patients with missing data will be exclude for the analysis.

The analysis of the safety of the surgical technique will be performed on the entire sample. Oncological safety and satisfaction assessment will be analyzed by groups: oncological patients and patients with a RRM.

### Monitoring

The monitoring committee is made up of medical and radiation oncologists, pathologists, breast surgeons, and radiologists. This committee evaluates the inclusion of each patient in the study. Surgical complications are assessed by the unit's surgical committee.

### Harms

In our study an adverse event will be defined as any untoward medical occurrence in a subject without regard to the possibility of a causal relationship. Adverse events will be collected after the subject has provided consent and enrolled in the study.

### Ethics and dissemination

The study has been approved by our hospital's research ethics committee (Ethics Committee of A Coruña-Ferrol. Spain. Prospectively registered on 20/Jul/2020. Registration code: 2020/95). All included patients will sign a specific informed consent form for the study. This consent form will be delivered by the surgeon in the preoperative consultation during which the patient may ask questions. It is estimated that the preliminary data will be published 2–3 years after

the study has begun and that the final data will be divulged at 6–7 years. The data obtained during this study will be published in indexed journals.

## Consent or assent

Consent will be given to the patient during the second consultation with the surgical specialist. In this consultation the intervention and the possibilities of participating in the study will be explained. in the same way, the doubts that the patient may have will be clarified. Later, the patient will talk with the nurse case manager, who will also explain the surgical technique and the results.

After this consultation, if the patient decides to participate in the study, she will sign the informed consent.

The patients are free to leave the study at any time, for this they will only have to request the consent of revocation.

## Confidentiality

All study-related information will be stored securely at the study site. All participant information will be stored in locked file cabinets in areas with limited access. All laboratory specimens, reports, data collection, process, and administrative forms will be identified by a coded ID [*identification*] number only to maintain participant confidentiality. All records that contain names or other personal identifiers, such as locator forms and informed consent forms, will be stored separately from study records identified by code number. All local databases will be secured with password-protected access systems. Forms, lists, logbooks, appointment books, and any other listings that link participant ID numbers to other identifying information will be stored in a separate, locked file in an area with limited access.

## Ancillary and post-trial care

The patients will continue to be followed up in the breast unit and the optimal treatment indicated at all times will be provided.

## Trial results

The first results will be published 3 months after including the last patient in the study, so we can assess the main result.

## Authorship

The main investigators and all surgeons who have included patients in the study will be included in the publications.

## Reproducible research

In the first publication we will present our surgical technique and the study protocol. in this way the study can be reproduced.

# Discussion

The PreQ-20 study will analyze the impact of PBR on 3 separate measures: safety, quality of life and cosmetic sequelae. Unlike other studies that analyzed these 3 measures jointly for women with breast cancer and RRM, our study will individualize the results for each of these 2 patient groups. This differentiation is necessary from the methodological point of view, given

that the 2 patient groups have separate characteristics that modify the results and its interpretation, especially in 4 contexts of the research. The first is in regard to the assessment of RGT, given that various studies have shown that its incidence is higher in RRM due to it being a more conservative mastectomy. The second context refers to the incidence of primary tumors in RGT. This fact requires differentiation between patients with cancer and those with a high risk because the latter have a higher likelihood of a primary carcinoma, especially carriers of the BRCA1 mutation. Third, the satisfaction and quality of life requires an independent assessment for each group because women at high risk are healthy, and therefore their expectations and experiences are very different from women with a diagnosis and treatment for breast cancer. Additionally, the physical and sexual wellbeing of patients with cancer is determined by the effects of the adjuvant therapy (hormone therapy, chemotherapy, radiation therapy) and are therefore experienced in different conditions and connotations compared with women at high risk. Lastly, there is a fourth differentiating element between the two groups that focuses on cosmetic sequelae. It is foreseeable that patients with cancer experience a higher incidence of cosmetic sequelae given that the implementation of radiation adjuvant therapy or salvage surgery (nipple excision, subcutaneous fat excision) result in a higher likelihood of cosmetic deterioration. These differences determine that the results of the PBR assessment should be differentiated for these 2 patient groups.

The safety assessment of PBR in women with breast cancer requires a minimum follow-up of 5 years. These results will therefore be published 6–7 years after the start of the study. Another secondary objective of the PreQ-20 study is the identification of risk factors for local recurrence after PBR and especially their link with ductal carcinoma *in situ* and the tumor's biological profile. Another objective will be to detect the onset of primary carcinoma in women with RRM, especially in carriers of the BRCA1 mutation. In our community, 45% of patients with the BRCA1 mutation have an ancestral mutation characteristic of Spain (R71G BRCA1) [24], which has shown greater lethality by promoting a high incidence of triple-negative tumors.

The assessment of RGT with MRI at 1 year of surgery helps determine the incidence rate while simultaneously providing information on the criteria for its assessment and the decision-making process for its follow-up or excision. This new modality for postmastectomy assessment starts a new culture in controlling and following-up sparing mastectomies aimed at the search for RGT.

The quality-of-life assessment in the PreQ-20 study will be conducted using a preoperative survey and 2 postoperative surveys (1 and 5 years after surgery). This methodology differs from most previous studies that only performed 1 postoperative assessment. The comparison between the preoperative and postoperative surveys will help identify those patients who specifically improve or worsen in their satisfaction and wellbeing, which will enable the identification of patient profiles with a higher likelihood of expected improvement or worsening in their satisfaction or wellbeing. By this method, individualized information processes can be established for patients to improve the joint decision-making process. This type of study has already been employed by the authors with women who underwent oncoplastic surgery to identify groups with a greater or lesser benefit for oncoreductive mammoplasty [25].

Lastly, the assessment of cosmetic sequelae will be performed through clinical examinations during the patients' follow-up in the doctor's office. Most of these sequelae will occur during the first 2 years. The PreQ-20 study will analyze their incidence, origin and, especially, the decisions taken to improve them. The study's prospective design will allow for a more specific assessment of these sequelae, given that the patients will be reviewed in the doctor's office over the course of at least 5 years.

## Conclusion

The PreQ-20 study's main objective is to assess the safety of PBR in terms of postsurgical complications and loss of implants. The results of this study will be differentiated for the groups of women with breast cancer and women with RRM. This differentiation will allow us to analyze the impact of this procedure independently in the contexts of cancer and risk reduction. The secondary objectives of the PreQ-20 study are focused on the analysis of risk factors for local recurrence in women with breast cancer and on identifying the patient profiles with foreseeable changes in their postoperative satisfaction and wellbeing.

## Supporting information

**S1 Checklist. SPIRIT 2013 checklist: Recommended items to address in a clinical trial protocol and related documents***.
(DOCX)

**S1 File.**
(DOCX)

## Acknowledgments

The authors would like to thank Samuel González of the Research Support Unit for his management of administrative paperwork for this project with the hospital's Ethics Committee and Management.

## Author Contributions

**Data curation:** Benigno Acea-Nebril, Alejandra García-Novoa.

**Formal analysis:** Benigno Acea-Nebril, Alejandra García-Novoa.

**Investigation:** Benigno Acea-Nebril, Alejandra García-Novoa.

**Methodology:** Benigno Acea-Nebril, Alejandra García-Novoa, Lourdes García Jiménez.

**Supervision:** Benigno Acea-Nebril.

**Validation:** Benigno Acea-Nebril.

**Writing – original draft:** Benigno Acea-Nebril.

**Writing – review & editing:** Alejandra García-Novoa.

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
