## [Decision Letter · Decision Letter 0]

14 Oct 2021

PONE-D-21-20755The PreQ-20 TRIAL, a prospective cohort study of the oncologic safety, quality of life and cosmetic outcomes of patients undergoing prepectoral breast reconstructionPLOS ONE

Dear Dr. García Novoa,

Thank you for submitting your manuscript to PLOS ONE. After careful consideration, we feel that it has merit but does not fully meet PLOS ONE’s publication criteria as it currently stands. Therefore, we invite you to submit a revised version of the manuscript that addresses the points raised during the review process.

We look forward to receiving your revised manuscript.

Kind regards,

Joon Jeong

Academic Editor

PLOS ONE

Journal Requirements:

3. Thank you for submitting your clinical trial to PLOS ONE and for providing the name of the registry and the registration number. The information in the registry entry suggests that your trial was registered after patient recruitment began. PLOS ONE strongly encourages authors to register all trials before recruiting the first participant in a study.

1) your reasons for your delay in registering this study (after enrolment of participants started);

2) confirmation that all related trials are registered by stating: “The authors confirm that all ongoing and related trials for this drug/intervention are registered”.

N/A 

6. Please amend your authorship list in your manuscript file to include author Lourdes García Jiménez.

7. Please amend your list of authors on the manuscript to ensure that each author is linked to an affiliation. Authors’ affiliations should reflect the institution where the work was done (if authors moved subsequently, you can also list the new affiliation stating “current affiliation:….” as necessary).

8. Please amend your manuscript to include your abstract after the title page.

Reviewers' comments:

Reviewer's Responses to Questions

**Comments to the Author**

1. Does the manuscript provide a valid rationale for the proposed study, with clearly identified and justified research questions?

Reviewer #1: Partly

Reviewer #2: No

2. Is the protocol technically sound and planned in a manner that will lead to a meaningful outcome and allow testing the stated hypotheses?

Reviewer #1: Partly

Reviewer #2: Partly

3. Is the methodology feasible and described in sufficient detail to allow the work to be replicable?

Reviewer #1: Yes

Reviewer #2: No

4. Have the authors described where all data underlying the findings will be made available when the study is complete?

Reviewer #1: No

Reviewer #2: Yes

5. Is the manuscript presented in an intelligible fashion and written in standard English?

Reviewer #1: Yes

Reviewer #2: Yes

6. Review Comments to the Author

You may also provide optional suggestions and comments to authors that they might find helpful in planning their study.

Reviewer #1: The PreQ-20 Trial is designed as a prospective cohort (a single arm) study to evaluate the safety (postoperative and oncologic safety, quality of life, and cosmetic outcomes of prepectoral breast reconstruction in patients who will receive skin-sparing mastectomy or nipple-sparing mastectomy. However, I have some concerns regarding the Study protocol and will be grateful if the authors could clarify these.

Abstract

Methods

1) The authors calculated the sample size based on the loss of implant. Therefore, only “loss of implant” is considered a primary endpoint, and the remaining postoperative complications should be considered secondary endpoints.

2) Please describe the statistical method for calculating the sample size.

3) Please clarify that this is a single-arm study.

Discussion

4) Isn’t “PreQ-20” right instead of “BreastQ-20”?

Manuscript

p.3 Table

5) Key inclusion and exclusion criteria: Patients who need a mastectomy for breast cancer or reducing risk: I would recommend changing “mastectomy” to “Skin-sparing mastectomy / Nipple-sparing mastectomy

Introduction

6) The introduction is redundant and needs to be revised more smoothly. In addition, it is necessary to clarify the purpose for which this study should be conducted.

Methods

7) Eligibility criteria: Women at high risk for breast cancer are included in this study. Is it correct to include benign breast tumors in the exclusion criteria?

8) Please clarify the definition of post-operative complications, including loss of implant, postoperative bleeding, skin necrosis, necrosis of the nipple-areola complex, seroma. and infection. Please describe how to evaluate these outcomes.

9) If there is residual glandular tissue on follow-up breast MRI, is it surgically removed or followed? Please clarify this.

10) Most of the results are measured at baseline, 12-18 months after surgery, and 5 years after surgery. Is there no follow-up between 12-18 months after surgery and 5 years after surgery? It needs to be clarified on this.

11) Please provide the criteria for “drop out” and enrollment/follow-up period.

Discussion

12) The authors said that the strength of this study is to evaluate safety in high risk for cancer/cancer groups separately. Is it enrolling 80 participants per group? If there are 80 people in total, is the ratio of the two groups preset?

13) QoL is measured at baseline, 12-18 months, and 60 months after surgery, but this is evaluated at different time points within one PBR group. It is thought that it is more important to compare with RBR or non-reconstruction populations, and it is necessary to discuss this.

Reviewer #2: 1. The authors introduced a new prospective study protocol(PreQ-20 Trial) for the oncologic safety, QoL, and aesthetic outcomes of prepectoral breast reconstruction. The followings are comments and criticisms.

2. First of all, it is not clear what the purpose of this study is. Is this a study comparing the differences in safety and patient satisfaction between the PBR and the RBR? Or is it a comparison of the results of the PBR between the cancer patients and the high-risk patients? If the purpose of this study is the former, this study should be re-designed with the RBR patient group as a control group. This might be a critical issue of this study. If the main purpose is the latter, the title should be changed as follows; “The PreQ-20 Trial, a prospective cohort study of the oncologic safety, quality of life and cosmetic outcomes of Patients WITH BREAST CANCER AND HIGH RISK undergoing prepectoral breast reconstruction”.

3. The authors included both SSM and NSM as study subjects. However, there is inevitably a big difference between patient satisfaction and cosmetic sequelae in SSM and NSM. No matter how good the symmetry of volume and shape is, the presence or absence of papilla has no choice but to influence satisfaction significantly. If both SSM and NSM are included, both methods should be equally assigned to the cancer patient and the high-risk groups through stratification. In addition, if the stratified extraction method is applied, the overall sample size should be larger than now.

4. Post-mastectomy radiation therapy(PMRT) and chemotherapy are performed only in cancer patients. PMRT is known to increase the rate of implant loss and capsular contracture, and chemotherapy is known to increase the incidence of postoperative infection or seroma. Therefore, the difference in outcome between cancer patients and high-risk patients is inevitably significant.

5. As for the implant type, the polyurethane(PU) implant is a very controversial prosthesis. The PU implant was taken off the worldwide market in 1991 but was later reintroduced. Studies have shown potential toxicity with polyurethane, and our studies have shown that it tends to disintegrate. Because polyurethane implants have the CE mark (the European Union stamp of approval), they can be marketed in the European Union.

The PU implants have not been reintroduced in the US since their discontinuation in 1991. They have not been used in North America, including Canada, these days. Also, they are not welcomed by the Ibero-Latin American Plastic Surgeons(FILACP), and the percentage of use of the implants is less than 5%. (Cir. Plást. Iberolatinoam. 46(2), 2020, 125-140). PU implant is also low in use in Asia and has been taken off from the market in several countries, including South Korea.

The PU implant's association with a rare malignant tumor called BIA-ALCL has also been reported. (Aesthet Surg J 39(S1);2019, S49–S54/ Aesthet Surg J 40(8);2020, 838-846 / Plast Reconstr Surg 143; 2019, 30S-40S.) Therefore, their use is no longer recommended in Australia and New Zealand.

Of course, there might be reflections, but I don't think it's appropriate for the PU implant to be used for safety assessment after breast cancer surgery.

7. PLOS authors have the option to publish the peer review history of their article (what does this mean?). If published, this will include your full peer review and any attached files.

Reviewer #1: No

Reviewer #2: No

---

## [Author Response · Author response to Decision Letter 0]

13 Nov 2021

Journal Requirements:

1) your reasons for your delay in registering this study (after enrolment of participants started)

The study began when the ethics committee of the Hospital made the preliminary approval of the study. However, we had to wait for the final approval report from the committee. Additionally, we first registered on the clinicaltrials.gov website.

Finally, we have managed the publication of the protocol in your journal.

We add a phrase in methods.

2) confirmation that all related trials are registered by stating: “The authors confirm that all ongoing and related trials for this drug/intervention are registered”.

We add the phrase in methods.

Our study has no funding. The Breast Unit is located in a Public Hospital and all the materials used in the surgery are financed by the public health.

 We do not have funders.

No authors received any salary from funders.

We add this phrase at protocol.

The authors believe that this means that they will provide payment information once the article is accepted.

6. Please amend your authorship list in your manuscript file to include author Lourdes García Jiménez.

We add the author.

7. Please amend your list of authors on the manuscript to ensure that each author is linked to an affiliation. Authors’ affiliations should reflect the institution where the work was done (if authors moved subsequently, you can also list the new affiliation stating “current affiliation:….” as necessary).

We add the affiliation.

8. Please amend your manuscript to include your abstract after the title page.

We add an abstract.

Reviewers' comments:

Comments to the Author

1. Does the manuscript provide a valid rationale for the proposed study, with clearly identified and justified research questions?

Reviewer #1: Partly

Reviewer #2: No

2. Is the protocol technically sound and planned in a manner that will lead to a meaningful outcome and allow testing the stated hypotheses?

Reviewer #1: Partly

Reviewer #2: Partly

3. Is the methodology feasible and described in sufficient detail to allow the work to be replicable?

Reviewer #1: Yes

Reviewer #2: No

4. Have the authors described where all data underlying the findings will be made available when the study is complete?

Reviewer #1: No

Reviewer #2: Yes

5. Is the manuscript presented in an intelligible fashion and written in standard English?

Reviewer #1: Yes

Reviewer #2: Yes

6. Review Comments to the Author

You may also provide optional suggestions and comments to authors that they might find helpful in planning their study.

Reviewer #1: The PreQ-20 Trial is designed as a prospective cohort (a single arm) study to evaluate the safety (postoperative and oncologic safety, quality of life, and cosmetic outcomes of prepectoral breast reconstruction in patients who will receive skin-sparing mastectomy or nipple-sparing mastectomy. However, I have some concerns regarding the Study protocol and will be grateful if the authors could clarify these.

Abstract

Methods

1) The authors calculated the sample size based on the loss of implant. Therefore, only “loss of implant” is considered a primary endpoint, and the remaining postoperative complications should be considered secondary endpoints.

The reviewer makes an approach that we consider correct. Our main endpoint, from which the sample size has been calculated, is the loss of implants. Therefore, we accept the reviewer's suggestion and put the rest of the complications as secondary endpoints.

2) Please describe the statistical method for calculating the sample size.

We use the sample size calculation formula for proportions. Our population is 250 thousand inhabitants. We assume that we can loss 10% of patient during follow-up. Our study will assume a 5% loss of implants. Therefore, with a 95% confidence level (95% CI), an accuracy of 5%, we need 81 mastectomies.

3) Please clarify that this is a single-arm study.

Yes, our study is a prospective single-arm study, in which all women included in the analysis will undergo a mastectomy with immediate reconstruction with a polyurethane implant in the prepectoral location. We add the phrase "single arm" in the study design section to improve the reader's understanding of our study

Discussion

4) Isn’t “PreQ-20” right instead of “BreastQ-20”?

We agree with the reviewer. We correct in the discussion, changing BreastQ 20 for PreQ 20.

Manuscript

p.3 Table

5) Key inclusion and exclusion criteria: Patients who need a mastectomy for breast cancer or reducing risk: I would recommend changing “mastectomy” to “Skin-sparing mastectomy / Nipple-sparing mastectomy

We accept the recommendation and make the modification in the text.

Introduction

6) The introduction is redundant and needs to be revised more smoothly. In addition, it is necessary to clarify the purpose for which this study should be conducted.

Thanks to the suggestion of the reviewer, we have optimized the introduction. We have summarized the points with incomplete analysis about the conservative mastectomies with immediate prepectoral reconstruction that motivated us to carry out this study. We consider that the literature published to date lacks long-term follow-up to assess the safety of the technique in terms of surgical complications and oncological safety. Furthermore, we do not know the evolution of the reconstruction over time. That is why we carried out this prospective study in which we intend to analyze these aspects with a long-term follow-up.

Methods

7) Eligibility criteria: Women at high risk for breast cancer are included in this study. Is it correct to include benign breast tumors in the exclusion criteria?

We understand the question postured by the reviewer, since cancer patients have different clinical and psychological characteristics from women who underwent a risk-reducing mastectomy. However, as our main objective is to determine the safety of the surgical technique in terms of implant losses, we consider the joint analysis of both patients is adequate. For this section we will analyze whether cancer treatments (chemotherapy and / or radiotherapy) increase the risk of surgical complications and loss of implants. Similarly, for the analysis of residual glandular tissue, the risk reduction and oncological groups can be grouped, since the amount of residual tissue assesses the quality of the mastectomy and the experience of the surgeon regardless of the diagnosis for which the mastectomy was performed. Subsequently, for the assessment of oncological safety and satisfaction of the surgery, an analysis by groups will be carried out, since these variables could be influenced by the criterion by which the mastectomy is performed.

To explain this, and avoid confusion for the reader, we add in the statistical analysis section the phrase: “The analysis of the safety of the surgical technique will be performed on the entire sample. Oncological safety and satisfaction assessment will be analyzed by groups: oncological patients and patients with a risk-reducing mastectomy.”

The reviewer's opinion of benign breast tumors is adequate. Therefore, we decided to remove this from the exclusion criteria.

8) Please clarify the definition of post-operative complications, including loss of implant, postoperative bleeding, skin necrosis, necrosis of the nipple-areola complex, seroma and infection. Please describe how to evaluate these outcomes.

At the reviewer's suggestion, these definitions are included in the outcomes section.

- Implant loss is defined as the need to remove the prosthesis for any cause such as strusion, infection or necrosis.

- Postoperative bleeding is defined as the appearance of any amount of bleeding that modifies the course of the first 7 postoperative days. Among these modifications we find: the need to prolong admission due to blood drainage and the need for reoperation due to bleeding or hematoma. Superficial ecchymoses that do not modify the immediate postoperative period are not included

- Skin necrosis is defined as the appearance of areas of skin without vascularization that condition cell death. Necrosis that appeared during the first 3 months after surgery that conditioned some action by the surgeon are included.

- Within the necrosis of the nipple areola complex, only deep necroses that required some surgical interventions are included.

- Seroma is defined as the accumulation of periprosthetic fluid that required maintaining the drain for more than 10 days or placing a new one (ultrasound-guided or at operating room).

- An infection is defined as the need for antibiotics in the first 30 postoperative days. The antibiotic prophylaxis used (Cefazolin the first 24 hours) is excluded.

9) If there is residual glandular tissue on follow-up breast MRI, is it surgically removed or followed? Please clarify this.

The decision to remove the residual glandular tissue will be made jointly between the patient and the surgeon, considering the amount of residual tissue, the vital status of the patient and the risk of a relapse or new tumor. Women with residual glandular tissue who do not undergo a reoperation will be followed up with an annual ultrasound and / or MRI. We add a sentence with this clarification in the section Participants' timeline.

10) Most of the results are measured at baseline, 12-18 months after surgery, and 5 years after surgery. Is there no follow-up between 12-18 months after surgery and 5 years after surgery? It needs to be clarified on this.

All patients with breast cancer will be followed up in the Breast Unit, both by the surgeon and by the medical oncologist, every 3-6 months. Patients with a risk-reducing mastectomy will be evaluated annually by the surgeon. In these reviews, oncological and cosmetic events will be evaluated. However, the postoperative breast Q will only be delivered one year and 5 years after completing the treatments. We add this clarification in the Participant's timeline section.

11) Please provide the criteria for “drop out” and enrollment/follow-up period.

With the phrase "drop out" we mean that by coordinating the study visits with the visits scheduled in the consultation for the follow-up of their disease, patients do not usually leave the study. We understand that the expression can be confusing, so we modify it by: “…patients do not usually leave the study”.

Discussion

12) The authors said that the strength of this study is to evaluate safety in high risk for cancer/cancer groups separately. Is it enrolling 80 participants per group? If there are 80 people in total, is the ratio of the two groups preset?

The sample size was calculated to determine the incidence of implant loss after a mastectomy with prepectoral reconstruction. For this, we include in the analysis all the mastectomies performed, regardless of the reason for which the mastectomy is performed (cancer or risk reduction). There are parts of the study (satisfaction for example) in which we will carry out the analysis separated by groups. We understand that this is a limitation of our study, since the precision of the results to analyze these data in which the sample size may be small is inadequate. That is why, once we complete the sample size, we intend to continue including patients in the study to later propose the analysis of these variables with greater precision and power. For future studies we will calculate the sample size according to each group.

13) QoL is measured at baseline, 12-18 months, and 60 months after surgery, but this is evaluated at different time points within one PBR group. It is thought that it is more important to compare with RBR or non-reconstruction populations, and it is necessary to discuss this.

As the reviewer comments, the study will carry out the satisfaction questionnaire (Breast Q) before the intervention, one year after finishing the treatments and after 5 years. The preoperative questionnaire allows us to know the baseline characteristics of the patients and thus be able to establish improvements or worsening after the intervention. The purpose of obtaining the questionnaire again at 5 years is to assess the deterioration of reconstruction over time and the impact it has on women's satisfaction. Furthermore, the authors believe that the psychology of women after the treatments or 5 years later is different, and therefore, the assessment of reconstruction may be different.

We agree with the reviewer that comparing patient satisfaction with another type of reconstruction (retropectoral for example) is an interesting study, however it is not the objective of this study, which aims to describe and analyze the characteristics, complications and evolution of the most used reconstruction currently in our unit.

In future studies the authors could perform a comparative analysis of both types of reconstruction. In the discussion of our results, we will make comparisons with other series with different types of reconstruction.

Reviewer #2: 1. The authors introduced a new prospective study protocol (PreQ-20 Trial) for the oncologic safety, QoL, and aesthetic outcomes of prepectoral breast reconstruction. The followings are comments and criticisms.

2. First of all, it is not clear what the purpose of this study is. Is this a study comparing the differences in safety and patient satisfaction between the PBR and the RBR? Or is it a comparison of the results of the PBR between the cancer patients and the high-risk patients? If the purpose of this study is the former, this study should be re-designed with the RBR patient group as a control group. This might be a critical issue of this study. If the main purpose is the latter, the title should be changed as follows; “The PreQ-20 Trial, a prospective cohort study of the oncologic safety, quality of life and cosmetic outcomes of Patients WITH BREAST CANCER AND HIGH RISK undergoing prepectoral breast reconstruction”.

PREQ-20 TRIAL is a prospective study that aims to describe and analyze the safety of the surgical technique, oncological safety and the satisfaction of women. PREQ-20 TRIAL is not a clinical trial or a comparative study. Therefore, we do not have two groups to compare. The main analysis, from which the sample size has been calculated, is to determine the safety of the technique in terms of implant loss. For the oncological safety and satisfaction analysis, we will analyze the risk reduction and oncological groups separately, but the main intention is not to compare these groups.

The authors believe that it is very appropriate to carry out a comparative study with other reconstructive techniques such as the retropectoral, but currently in the breast unit, the retropectoral technique is limited to a few selected cases, so we could not carry out a comparative prospective study. However, as we have commented to the other reviewer, in the future we could carry out a comparative study with other techniques.

3. The authors included both SSM and NSM as study subjects. However, there is inevitably a big difference between patient satisfaction and cosmetic sequelae in SSM and NSM. No matter how good the symmetry of volume and shape is, the presence or absence of papilla has no choice but to influence satisfaction significantly. If both SSM and NSM are included, both methods should be equally assigned to the cancer patient and the high-risk groups through stratification. In addition, if the stratified extraction method is applied, the overall sample size should be larger than now.

Some studies have compared satisfaction between SSM and NSM. This aspect will be analyzed in our study, in order to confirm whether women who preserve the nipple really have greater satisfaction with its reconstruction. Women in whom the nipple has not been preserved are offered to reconstruct it through outpatient surgery. Those in whom the areola has been removed, the tattoo of it is offered.

The reviewer proposes stratification according to nipple preservation. The authors believe that this is not possible. In the first place, because the main objective is to assess the safety of the technique in terms of implant loss. Therefore, satisfaction according to the preservation of the nipple will simply be a variable to be analyzed. On the other hand, the decision to preserve the nipple depends on 2 fundamental aspects that make its stratification difficult. In cancer patients, the nipple can only be preserved in those who her nipple is not invaded by cancer. The other aspect is that the decision on the type of incision (type IV, inframammary fold) will depend on the volume and ptosis of the breast.

Our study will provide data on the satisfaction of those women who preserved the nipple in each group (risk reduction and cancer), but they do not be stratified. 

4. Post-mastectomy radiation therapy (PMRT) and chemotherapy are performed only in cancer patients. PMRT is known to increase the rate of implant loss and capsular contracture, and chemotherapy is known to increase the incidence of postoperative infection or seroma. Therefore, the difference in outcome between cancer patients and high-risk patients is inevitably significant.

We agree with the reviewer that some studies had proven that radiation and chemotherapy could increase complications related with implant. However, there are no studies that have confirmed these data with polyurethane implants in the prepectoral position. That is why the authors want to analyze whether these treatments influence post-surgical complications.

5. As for the implant type, the polyurethane(PU) implant is a very controversial prosthesis. The PU implant was taken off the worldwide market in 1991 but was later reintroduced. Studies have shown potential toxicity with polyurethane, and our studies have shown that it tends to disintegrate. Because polyurethane implants have the CE mark (the European Union stamp of approval), they can be marketed in the European Union.

The PU implants have not been reintroduced in the US since their discontinuation in 1991. They have not been used in North America, including Canada, these days. Also, they are not welcomed by the Ibero-Latin American Plastic Surgeons(FILACP), and the percentage of use of the implants is less than 5%. (Cir. Plást. Iberolatinoam. 46(2), 2020, 125-140). PU implant is also low in use in Asia and has been taken off from the market in several countries, including South Korea.

The PU implant's association with a rare malignant tumor called BIA-ALCL has also been reported. (Aesthet Surg J 39(S1);2019, S49–S54/ Aesthet Surg J 40(8);2020, 838-846 / Plast Reconstr Surg 143; 2019, 30S-40S.) Therefore, their use is no longer recommended in Australia and New Zealand.

Of course, there might be reflections, but I don't think it's appropriate for the PU implant to be used for safety assessment after breast cancer surgery.

The reflection made by the reviewer is very interesting. As he says, in the 90s there was a campaign against polyurethane implants and in fact the FDA banned their use. However, this alert was removed as they failed to confirm the association between polyurethane and cancer. In fact, some experimental studies confirmed that the amount of the carcinogenic metabolite generated from the degradation of polyurethane is not capable of producing cancer in humans (Hester TR et al. Plas Reconstr Surg. 1997)

However, one of the main motivations for this study is to confirm the safety of polyurethane, since currently in Spain and a large part of Europe are the most widely used implants for prepectoral reconstruction.

We welcome the reviewer's comment and we will do a long-term, comprehensive follow-up to determine the safety of these implants in our patients.

7. PLOS authors have the option to publish the peer review history of their article (what does this mean?). If published, this will include your full peer review and any attached files.

Do you want your identity to be public for this peer review? For information about this choice, including consent withdrawal, please see our Privacy Policy.

Reviewer #1: No

Reviewer #2: No

---

## [Decision Letter · Decision Letter 1]

23 Dec 2021

PONE-D-21-20755R1The PreQ-20 TRIAL, a prospective cohort study of the oncologic safety, quality of life and cosmetic outcomes of patients undergoing prepectoral breast reconstructionPLOS ONE

Dear Dr. García Novoa,

Thank you for submitting your manuscript to PLOS ONE. After careful consideration, we feel that it has merit but does not fully meet PLOS ONE’s publication criteria as it currently stands. Therefore, we invite you to submit a revised version of the manuscript that addresses the points raised during the review process.

We look forward to receiving your revised manuscript.

Kind regards,

Joon Jeong

Academic Editor

PLOS ONE

Reviewers' comments:

Reviewer's Responses to Questions

**Comments to the Author**

1. Does the manuscript provide a valid rationale for the proposed study, with clearly identified and justified research questions?

Reviewer #1: Yes

Reviewer #2: Partly

2. Is the protocol technically sound and planned in a manner that will lead to a meaningful outcome and allow testing the stated hypotheses?

Reviewer #1: Yes

Reviewer #2: Yes

3. Is the methodology feasible and described in sufficient detail to allow the work to be replicable?

Reviewer #1: Yes

Reviewer #2: Yes

4. Have the authors described where all data underlying the findings will be made available when the study is complete?

Reviewer #1: Yes

Reviewer #2: Yes

5. Is the manuscript presented in an intelligible fashion and written in standard English?

Reviewer #1: Yes

Reviewer #2: Yes

6. Review Comments to the Author

You may also provide optional suggestions and comments to authors that they might find helpful in planning their study.

Reviewer #1: The authors faithfully responded to the reviewer's comments and revised the manuscript. However, it is necessary to change the "Breast Q-20 trial" to the "PreQ-20 trial" in the discussion of Abstract.

Reviewer #2: There are still some points that are difficult to agree with, but there seems to be no problem conducting clinical trials to assess the safety of the prepectoral implant based breast reconstruction as the current protocol.

7. PLOS authors have the option to publish the peer review history of their article (what does this mean?). If published, this will include your full peer review and any attached files.

Reviewer #1: No

Reviewer #2: No

---

## [Author Response · Author response to Decision Letter 1]

11 Jan 2022

Comments to the Author

Reviewer #1: 

The authors faithfully responded to the reviewer's comments and revised the manuscript. However, it is necessary to change the "Breast Q-20 trial" to the "PreQ-20 trial" in the discussion of Abstract.

We appreciate the reviewer's comment. We apologize for the mistake in the abstract discussion, and we correct it.

Reviewer #2: 

There are still some points that are difficult to agree with, but there seems to be no problem conducting clinical trials to assess the safety of the prepectoral implant based breast reconstruction as the current protocol.

Thanks for the reviewer's comments. We also appreciate that the reviewer accepts the approach of the protocol.

---

## [Decision Letter · Decision Letter 2]

20 May 2022

The PreQ-20 TRIAL, a prospective cohort study of the oncologic safety, quality of life and cosmetic outcomes of patients undergoing prepectoral breast reconstruction

PONE-D-21-20755R2

Dear Dr. García Novoa,

We’re pleased to inform you that your manuscript has been judged scientifically suitable for publication and will be formally accepted for publication once it meets all outstanding technical requirements.

Kind regards,

Yann Benetreau, PhD

Division Editor (staff editor)

PLOS ONE

Additional Editor Comments (optional):

Reviewers' comments:

Reviewer's Responses to Questions

**Comments to the Author**

1. Does the manuscript provide a valid rationale for the proposed study, with clearly identified and justified research questions?

Reviewer #1: Yes

Reviewer #2: Yes

2. Is the protocol technically sound and planned in a manner that will lead to a meaningful outcome and allow testing the stated hypotheses?

Reviewer #1: Yes

Reviewer #2: Yes

3. Is the methodology feasible and described in sufficient detail to allow the work to be replicable?

Reviewer #1: Yes

Reviewer #2: Yes

4. Have the authors described where all data underlying the findings will be made available when the study is complete?

Reviewer #1: Yes

Reviewer #2: Yes

5. Is the manuscript presented in an intelligible fashion and written in standard English?

Reviewer #1: Yes

Reviewer #2: Yes

6. Review Comments to the Author

You may also provide optional suggestions and comments to authors that they might find helpful in planning their study.

Reviewer #1: The authors faithfully responded to the reviewer's comments and revised the manuscript. There seems to be no problem conducting clinical trial.

Reviewer #2: No additional comment this time except for the implant issue(polyurethane coated implant which is not widely accepted in the world)

7. PLOS authors have the option to publish the peer review history of their article (what does this mean?). If published, this will include your full peer review and any attached files.

Reviewer #1: No

Reviewer #2: No

---

## [Editor Report · Acceptance letter]

13 Jun 2022

PONE-D-21-20755R2 

The PreQ-20 TRIAL: a prospective cohort study of the oncologic safety, quality of life and cosmetic outcomes of patients undergoing prepectoral breast reconstruction 

Dear Dr. García Novoa:

I'm pleased to inform you that your manuscript has been deemed suitable for publication in PLOS ONE. Congratulations! Your manuscript is now with our production department. 

Kind regards, 

on behalf of

Dr. Yann Benetreau 

Staff Editor

PLOS ONE